# Papillomavirus-like Particles in Equine Medicine

**DOI:** 10.3390/v15020345

**Published:** 2023-01-25

**Authors:** Edmund K. Hainisch, Christoph Jindra, Reinhard Kirnbauer, Sabine Brandt

**Affiliations:** 1Research Group Oncology (RGO), Clinical Unit of Equine Surgery, Department for Companion Animals and Horses, Veterinary University, 1210 Vienna, Austria; 2Division of Molecular Oncology and Haematology, Karl Landsteiner University of Health Sciences, 3500 Krems an der Donau, Austria; 3Laboratory of Viral Oncology (LVO), Department of Dermatology, Medical University, 1090 Vienna, Austria

**Keywords:** papillomaviruses, horses, BPV1, BPV2, EcPV2, sarcoid, squamous cell carcinoma, virus-like particles, prophylactic vaccines

## Abstract

Papillomaviruses (PVs) are a family of small DNA tumor viruses that can induce benign lesions or cancer in vertebrates. The observation that animal PV capsid-proteins spontaneously self-assemble to empty, highly immunogenic virus-like particles (VLPs) has led to the establishment of vaccines that efficiently protect humans from specific PV infections and associated diseases. We provide an overview of PV-induced tumors in horses and other equids, discuss possible routes of PV transmission in equid species, and present recent developments aiming at introducing the PV VLP-based vaccine technology into equine medicine.

## 1. Introduction

The Papillomaviridae family comprises a high number of small non-enveloped viruses that are characterized by great genetic diversity yet adhere to common biological principles. All papillomaviruses (PV) consist of a capsid harboring a circular double-stranded DNA genome of up to 8 kb in length. The viral capsid is composed of 72 L1 protein pentamers termed capsomers, and a minimum of 12 L2 protein monomers [1]. The PV genome can be divided into an early (E), a late (L) and a non-coding long control region (LCR). The early (E) region codes for regulatory (e.g., E1, E2, E4) and transforming proteins (e.g., E5, E6, E7). The late (L) region encodes the major capsid protein L1, and the minor capsid protein L2. The LCR is located downstream of the L1 gene, and provides cis-responsive elements required for replication and transcription of the PV genome [2]. 

PVs can infect a wide variety of vertebrates, including humans. Early research in natural animal models has revealed their usual tropism for epithelial keratinocytes and their species-specificity [3]. Delta- (δ-)PVs are an exception to this general rule, in that they also or mainly infect dermal fibroblast and—perhaps for this reason— have a wider host range. This is best illustrated by bovine δ-PV types 1 and 2 (BPV1; BPV2), which not only infect cattle, but also other ungulates including horses [4,5]. Interestingly, there is also recent evidence of ovine PVs infecting cattle [6].

Depending on the respective presence and transforming potential of the viral proteins E6, E7, and E5, PVs can induce epithelial lesions ranging from benign papillomas to life-threatening cancers [5,7,8,9,10,11,12,13,14,15]. From the more than 200 human PV (HPV) types known today, 15 have been meanwhile recognized as carcinogenic, and hence are designated as high-risk HPVs (hrHPVs) [16]. The latter are the major causative agents of virtually all cervical cancers, the majority of anal and oropharyngeal cancers, and approximately 50% of vaginal, vulvar and penile squamous cell carcinomas (SCCs) in humans [17]. 

In horses, four PV types have been identified as (putative) tumorigenic PVs so far, i.e., BPV1, BPV2, BPV13 [4,18,19], and equine PV type 2 (EcPV2)—a Dyoiota-PV [20,21].

## 2. BPV-Induced Equine Sarcoids

Sarcoids are benign skin tumors affecting up to 12% of horses worldwide and other equid species such as donkeys, mules, or zebras [22]. Sarcoids can develop anywhere on the integument and are classified as occult, verrucous, nodular, fibroblastic, mixed, or malevolent lesions, according to their clinical appearance and gross morphology (Figure 1) [23]. 

Sarcoids constitute a dreaded horse disease, although they do not metastasize. First, lesions can develop at body sites that may restrict or compromise the use of affected animals as riding, draft, or breeding horses. Second, mild-type sarcoids can rapidly progress to more severe, multiple lesions upon accidental or iatrogenic trauma. This implies that invasive therapeutic interventions often result in tumors recrudescing in a more aggressive, multiple form [24,25]. Third, sarcoids also have a significant economic impact, as they considerably reduce the resale value of affected equids and are associated with possibly lifelong treatment costs [4,26]. As a result, sarcoids constitute a major dermatological reason for euthanasia [25].

First evidence of an etiological association of BPV1 and BPV2 with sarcoid disease was provided in the 1950s, when inoculation of equine skin with cell-free extract supernatant prepared from BPV1- or 2-positive cow warts led to the development of lesions that were clinically and morphologically indistinguishable from natural sarcoids [27]. With the advent of modern molecular-biological techniques, the causal association of BPV1 and BPV2 infection with sarcoid onset and progression was unequivocally demonstrated [26]. It is accepted today that transformation of infected cells is mediated by a concerted action of the BPV1, BPV2 (and probably BPV13) oncoproteins E5, E6, and E7. In cultured cells and cattle, the small hydrophobic E5 oncoprotein has been shown to transform infected cells via binding to the platelet-derived growth-factor-β receptor (PDGFβ-R), leading to activation of various protumoral kinases [28,29,30,31]. Although horse-specific data are still lacking, it is conceivable that E5 likewise binds to and activates PDGFβ-R in equid cells. Importantly, E5 also interferes with MHC I biosynthesis and transport to the cell surface, thus impairing the antigen-recognition machinery. [32]. In addition, bovine δ-PV E5 proteins were shown to impair innate immune-signaling pathways mediated by RIG-I-like receptors and cGAS-STING. [33,34]. E5 thus compromises the innate and adaptive immune-response to bovine δ-PVs. The E6 oncoprotein competitively binds to the focal-adhesion protein paxillin, thus compromising its function and enabling infected cells to grow in an anchorage-independent manner [14]. The E7 oncoprotein likely contributes to this growth characteristic, by inhibiting anoikis [35]. In addition, E7 acts as enhancer of E5- and E6-mediated cell transformation, e.g., by its ability to bind to p600, a 600-kDa retinoblastoma- and calmodulin-binding protein required for membrane morphogenesis and cell survival [31]. 

In 2013, Lunardi et al. reported the isolation of a new PV from a bovine papilloma in Brazil. Genetic characterization of this PV revealed a more than 90% identity with BPV1 and BPV2 on predicted amino-acid level. On these grounds, this PV, termed BPV13, was classified as δ-PV [18]. Interestingly, BPV13 DNA was also detected in sarcoids from two Brazilian horses [19]. Whilst in-depth analyses on the tumorigenic potential of BPV13 are still lacking, the close phylogenetic relationship with BPV1 and BPV2 suggests that BPV13 may represent a δ-PV type with similar transforming potential [18]. 

## 3. EcPV2-Associated Equine Squamous-Cell-Carcinoma 

Squamous cell carcinomas (SCCs) are malignant epithelial tumors that can arise from cutaneous and mucosal keratinocytes. In horses, SCCs predominantly develop at mucocutaneous junctions in the anogenital and ocular region. In addition, SCCs are commonly diagnosed in the head-and-neck (HN) region, i.e., the oral and/or sinonasal cavity, the larynx, and the pharynx, as well as unpigmented areas of the skin. At initial-tumor stages, lesions can present as whitish, benign plaques or papillomas that can progress to carcinoma in situ (CIS), and ultimately to invasive SCC [36,37] (Figure 2). 

Given the paucity of therapeutic options for equine tumor patients, treatment mainly consists of wide surgical excision of affected tissue. However, this is not always indicated or feasible, e.g., in the case of large, metastasizing lesions or inaccessible tumors, such as sinonasal, laryngeal or pharyngeal SCCs [21,36,38,39,40].

First evidence of a viral association with equine SCC was obtained in 2010, when Equus caballus PV type 2 (EcPV2) was isolated from an equine genital SCC and genetically characterized [20]. The EcPV2 genome revealed open reading frames (ORFs) for the viral genes E1, E2, E4, E6, E7, L2, and L1. Whilst the predicted E7 protein lacked the retinoblastoma-binding domain LXCXE, the predicted E6 protein sequence harbored a PDZ binding domain (XS/TXV/L), a typical feature of carcinogenic HPVs [41]. 

Subsequent PCR screening of equine SCCs by several groups revealed EcPV2 DNA in up to 100% of equine genital SCCs and precursor lesions [21]. In addition, a subset of equine HNSCCs scored positive for EcPV2 DNA [42,43,44,45], whilst ocular SCCs tested consistently negative for this virus [21]. Together with these findings, the low incidence of EcPV2 infection in tumor-free horses and the detection of viral transcripts in EcPV2 DNA-positive lesions [42,46,47] are highly suggestive of a causal association of EcPV2 with equine genital SCCs and a subset of HNSCCs.

## 4. Viral Transmission 

De novo infection by epitheliotropic hrHPVs is best studied for HPV16, the most commonly detected hrHPV type in HPV-associated human cancers [48]. Transmission mainly occurs by sexual contact, and requires epithelial microabrasions to allow infectious virus particles (virions) to gain access to basal stem-cell-like keratinocytes. The latter display the necessary surface molecules for virion attachment and endocytosis [48,49,50,51,52]. Following virus entry, the productive lifecycle of hrHPVs is tightly linked to the differentiation program of epithelial keratinocytes. PVs express their early genes in the basal and suprabasal layers, replicate their genomes in the differentiating spinous and granular layers, and finally express their late capsid genes in the squamous layer, where new virions are assembled and released via the desquamation of corneocytes [2]. Given the demonstrated restriction of EcPV2 infection to equine tumor epidermis [20,46], and the detection of putative virions from a subset of equine genital SCCs [45], it appears reasonable to assume that the entry and lifecycle of EcPV2 adheres to similar biological principles. However, the routes of EcPV2 transmission in equid populations have not been addressed so far. Consistent detection of a genetic “Icelandic” EcPV2 variant from SCCs affecting elderly Icelandic horses imported from Iceland to Austria in their youth, points to infection having occurred in Iceland and persisted for a long time in a latent state [45]. Foals and young individuals are curious and playful. They inspect the environment and their companions’ bodies by sniffing, licking, sucking, and biting. In addition, young stallions playfully practice mating on other horses of both sexes (EKH and SB: personal observation). It is speculated that this behavior is responsible for transmission of EcPV2 from infected to young healthy individuals. In addition, EcPV2 infection may be acquired by licking and nuzzling of contaminated surfaces. This is a likely scenario, since PV virions can persist for a long time in the environment without losing their infectivity [53]. 

Δ-papillomaviruses including BPV1, 2 and 13 differ from epitheliotropic PVs in that they also infect dermal fibroblasts [54]. In cattle, BPV1, 2 and 13 infections give rise to benign warts that usually regress spontaneously [18,54]. In the epidermal part of the lesions, infection is productive. In the dermal portion, however, fibroblasts harbor multiple viral episomes and do not support virion formation [54]. 

In horses, it was long time thought that BPV1/2 exclusively reside in fibroblasts in a non-productive episomal form, and that horses thus represent a dead-end host for these BPV types [55]. This theory pronouncedly hampered our understanding of BPV transmission to and within horse populations, and gave rise to speculations that de novo infection would possibly occur via viral DNA or infected fibroblasts. However, several studies refute these assumptions and the concept of an abortive BPV1/2-infection in equids. As early as 1972, Robl et al. reported on the successful intracranial infection of hamsters with native BPV1 virions, whilst heat-denatured virions failed to produce infection [56]. In 2008, first evidence for the putative presence of BPV1 virions in a subset of sarcoids was obtained by immunocapture PCR (IC/PCR) and IC/qPCR, using BPV1 L1-specific antibodies for virion trapping. Except for a severely affected horse, the quantitative approach revealed low amounts of putative intralesional virions [57]. This evidence was strengthened by Wilson et al., who visualized BPV1 virions in sarcoid tissue using electron microscopy [58]. Simultaneously, two independent groups demonstrated that BPV1 infection is not confined to the dermis, but also involves the epidermis, the accepted site of productive PV infection [59,60]. Furthermore, intradermal injection of horses with cow wart-derived BPV1 and BPV2 virions was shown to consistently result in pseudo-sarcoid development, whilst inoculation with BPV1-infected cells or naked viral genomes failed to induce visible skin reactions [61]. These observations were corroborated by Gobeil et al., who demonstrated that equine sarcoids are not inducible by an infectious cell line [62]. Together with co-stabling experiments resulting in healthy donkeys acquiring BPV1 infection from sarcoid-affected stablemates [63], these data show, in sum, that (1) BPV1/2 infection can be productive in horses, albeit at a low level, and that (2) infectious virions are required for de novo infection of equids. This latter point led to the assumption that induction of an immune response directed against the viral capsid should be able to prevent de novo infection by compromising virus entry.

Although the routes of natural BPV1/2-virion-transmission between equids are still a matter of debate, there are indications of direct contact, contaminated material/surfaces, and possibly insect vectors having a major role in BPV1/2-virion propagation [26,63,64,65,66,67,68]. For example, putative BPV1 virions were detected by IC/PCR from dandruff collected from the grooming kit of sarcoid-affected donkeys in Italy (SB, personal observation). 

## 5. Virus-like Particles as Prophylactic Horse-Vaccines 

The discovery that recombinantly expressed PV L1 (or L1 plus L2) capsid proteins spontaneously self-assemble to highly immunogenic virus capsids termed virus-like particles (VLPs) [69] initiated a revolution in PV-related prevention research. From this moment on, it seemed possible to generate vaccines for protection against PV infections and associated benign and malignant diseases. Given the usual species-specificity of PVs, first immunization studies were conducted with homologous VLPs in animal PV-models: Suzich et al. produced canine oral PV (COPV) VLPs and showed that vaccinated dogs, as well as dogs immunized with serum from the former, became resistant to experimental challenge with COPV [70]. Likewise, promising results were obtained with cottontail-rabbit PV (CRPV) VLPs in their natural host species [71,72], and BPV type 4 VLPs in calves [73]. Importantly, animal studies revealed that immunization with VLPs induced type-specific neutralizing antibodies that were necessary, but also sufficient to confer protection from challenge with respective PV-types [70,71,72,73,74]. To summarize, these data paved the way for the establishment of HPV type-specific VLPs, ultimately leading to the approval and commercialization of preventive HPV vaccines [75,76]. 

Given the high veterinary and economic relevance of sarcoids in horses and other equids, and efficient protection from HPV infection conferred by VLP-based vaccines in humans, efforts are being made to develop a sarcoid vaccine using the VLP technology. In a clinical Phase I trial involving 15 healthy adult horses, intramuscular immunization with 50, 100, and 150 µg of BPV1 L1 VLPs (Figure 3) in alum adjuvant on days 1, 28, and 168 was shown to be safe and highly immunogenic [77]. To address whether BPV1 L1-specific VLPs also have the potential to protect from BPV2 infection, the cross-neutralizing capacity of rabbit antisera induced by BPV1- and BPV2-L1-specific VLPs was analyzed. Importantly, antisera induced by either VLP vaccine were able to robustly (cross-) neutralize the heterologous as well as the homologous BPV-type, indicating that BPV1 and BPV2 are closely related serotypes. Hence, it was assumed that a monovalent BPV1 L1 VLP vaccine would be able to confer sufficient protection also from BPV2 infection [78]. 

In 1951, Olson and Cook succeeded in inducing pseudo-sarcoids in horses by intradermal inoculation with BPV1 virions. These lesions were macroscopically indistinguishable from natural sarcoids, but spontaneously regressed after several weeks [27]. Sixty years later, Hartl et al. successfully recapitulated this experiment in horses under more standardized conditions, i.e., by intradermal inoculation with defined concentrations of infectious BPV1-virions (5 x 10^5^ and 10^6^ virions per injection) (Figure 4) [61]. 

This pseudo-sarcoid model was subsequently used to assess the protective potential of BPV1 L1 VLPs (Figure 4) in a challenge study involving 21 sarcoid- and BPV1/2-free healthy horses. Fourteen individuals were immunized with 100 µg of BPV1 L1 VLPs in alum adjuvant on days 1 and 28, whilst seven control horses remained unvaccinated. On day 42, all 21 horses were intradermally inoculated with 10^6^ virions per injection at 10 sites on the left side of the neck. Importantly, 13/14 horses showed complete protection from BPV1 infection and associated pseudo-sarcoid formation. Only one 28-years-old individual developed tiny lesions (<2 mm in diameter) at several virion inoculation sites. These lesions regressed within several days. In contrast, non-vaccinated control horses developed nodular pseudo-sarcoids upon challenge with BPV1 virions at all injection sites. Lesions persisted from several weeks to months, and reached maximum diameters of 16 mm. These clinical data perfectly matched day-42 anti-BPV1 L1 antibody titers determined by pseudovirion (PsV) neutralization assay. These titers ranged between 6400 and 25,600 in the case of fully protected horses, but only reached 800 in the case of the partially protected individual. No BPV1 L1-specific antibodies were detected from sera of control horses [79].

In a second challenge-study conducted in an analogous manner, 14/21 horses were immunized with a bivalent vaccine containing 50 µg BPV1 L1 VLPs and 50 µg EcPV2 L1 VLPs. The rationale underlying this approach was to simultaneously assess (1) the in vivo potential of BPV1 VLPs to confer protection from BPV2, and (2) the safety and immunogenicity of EcPV2 VLPs. On day 42, all horses were challenged with BPV2 virions. Highly significant protection of vaccinated animals from BPV2-associated tumor formation was achieved (*p* = 0.0039). However, several vaccinated horses developed tiny lesions that reached maximum diameters of 5 mm and persisted for three weeks, on average. Measured day-42 anti-BPV1 L1 antibody titers were significantly lower compared to those determined during the BPV1 challenge-study and were exceeded by anti-EcPV2 L1-antibody titers. This finding is indicative for a suboptimal composition of the bivalent vaccine and the need for adjustment of respective antigen doses. Importantly, the obtained findings also showed that EcPV2 VLPs were safe and induced a robust type-specific antibody response in the horses [79]. 

An EcPV2 challenge was not envisaged in horses, due to ethical reasons. Instead, the protective potential of EcPV2 VLPs was addressed in an established surrogate mouse-model, using luciferase-expressing EcPV2 pseudo-virions (PsVs) for the challenge. In the first step, rabbits were immunized with EcPV2 VLPs, mutated (non-assembling) EcPV2 L1 protein, or BPV1 VLPs. Then, pre-immune and the respective immune sera were transferred to mice. Subsequently, the latter were intravaginally inoculated with EcPV2 PsVs, and protection from infection was assessed using bioluminescence imaging. Passive transfer of rabbit EcPV2-VLP immune-sera conferred complete protection from EcPV2 PsV infection. In contrast, no protection was achieved in mice transferred with antiserum raised against mutated EcPV2 L1 protein, heterologous BPV1 L1 VLP, or the respective pre-immune sera, confirming conformational dependence and type restriction of prophylactic EcPV2 VLP immunization [80]. 

To address the longevity of the protection conferred by BPV1 VLPs, the still available horses involved in the Phase I safety-and-immunogenicity trial (n = 7) were challenged with BPV1 virions (10^6^ virions per injection; 10× per horse) five years after their third (and last) immunization with 50, 100 or 150 µg of BPV1 VLPs. Three non-vaccinated horses were included in the challenge study as controls. Importantly, all vaccinated horses were completely protected from BPV infection and tumor formation, irrespective of VLP dosage, whilst control horses developed pseudo-sarcoids at all inoculation sites. This finding indicates that BPV1 VLP-mediated protection of horses is long-lasting. Complete protection was all the more remarkable, since it was achieved in horses revealing low-to-undetectable BPV1 L1-neutralizing antibody titers immediately before the challenge [81]. 

Immunization of humans with HPV VLPs was shown to elicit an antibody response that is up to 4 logs higher than the one induced by natural HPV infection [82]. Similarly, the antibody response to natural BPV1 infection in cattle is poor, and virtually non-existent in horses [54,83]. The remarkable protective efficacy of HPV VLPs is thought to be mediated by intramuscular administration of the vaccine (compared to natural infection, which is confined to the epithelia), which elicits a strong T-helper-cell-dependent B-cell response. This results in the generation of high-titer neutralizing antibodies and the establishment of B-cell memory [82,84]. Yet, there is growing evidence of cytotoxic T cells (CTLs) and effector memory T cells (CD8^+^ T_EM_) also having a major role in the establishment of VLP-mediated protection [85]. For example, immunization of healthy volunteers with HPV VLPs was shown to elicit an L1 protein-specific CTL response [86,87]. By analogy, immunization of horses with BPV1 VLPs may induce an L1-specific CD8^+^ T_EM_ response that may sustain vaccination-mediated protection from BPV1 infection against the background of decreasing antibody titers: CD8^+^ T_EM_ may migrate into the integument and become long-lived resident memory T cells (CD8^+^ T_RM_), constituting a powerful line of defense [85]. 

## 6. Virus-like Particles as Therapeutic Vaccines in Equid Tumor Management

As early as in the 1990s, the high immunogenicity of PV VLPs also raised the question of whether they may have therapeutic potential. To address this issue, Kirnbauer and colleagues infected calves via palatal inoculation with BPV4, thereby inducing papillomas. Then, half of animals were injected twice with BPV4 L1L2 VLPs, whilst the other calves served as controls. Although all VLP-treated calves seroconverted, vaccination had no significant therapeutic effect [73]. This finding is not surprising. Antibodies raised against L1 or L1L2 VLPs are capsid protein-specific and sterilizing, thus having the ability to bind virions and prevent their entry into the host cell. In established infection, however, PVs predominantly or solely express the viral early proteins that are not recognized by anti-L1 (-L2) antibodies [74,84,88,89]

BPV4 E7 fusion proteins were previously shown to accelerate regression of experimental palatal papillomas in calves [90]. Based on this finding, Ashrafi et al. generated chimeric BPV1 L1 VLPs also containing E7 peptides (CVLPs) as a potential therapeutic sarcoid vaccine. In a placebo-controlled trial, sarcoid-bearing donkeys received the vaccine or placebo (PBS) at days 0, 14, 35, 49, 70, 95 and 112 through intramuscular injection. Although a tendency towards regression and reduced progression of some of the CVLP-treated lesions was noted, no significant therapeutic effect could be reached [91]. Similar results were obtained in sarcoid-affected horses, where sarcoid growth rather than regression was observed in twelve CVLP-treated animals. In addition, only 5/12 sarcoid-patients seroconverted to the E7 component of the vaccine, as revealed by the detection of low-titer E7 antibodies [92].

## 7. Conclusions

As seen in humans, horses and other equids can develop different tumor diseases in the course of their lives. These include PV-induced lesions, i.e., benign sarcoids and malignant SCCs in the genital and the HN region [21,22]. In the early 1990s, in vitro generated PV L1 proteins were shown to spontaneously self-assemble to VLPs that proved safe and highly immunogenic in several animal PV-models. This discovery paved the way for the establishment and market authorization of VLP-based vaccines for the protection of humans against HPV infections and related diseases [75,93]. Given the protective efficacy of HPV VLPs, attempts were made to (re-)introduce the technology into veterinary medicine by developing BPV1 and EcPV2 L1 VLPs for the prevention of infection by these oncogenic virus types in equids.

Investigational vaccinations with BPV1 L1 VLPs were shown to be safe and highly immunogenic in horses [77], and their potential to completely protect from BPV1 infection and associated pseudo-sarcoid formation was unequivocally demonstrated [79]. In addition, immunization of horses with BPV1 L1 VLPs elicited antibodies that also cross-neutralized BPV2 [78,79]. Intramuscular administration of a bivalent BPV1 L1/EcPV2 L1 VLP vaccine was likewise safe, and induced a specific antibody response against both PV types [79]. The prophylactic potential of the EcPV2 L1 VLP vaccination was assessed in a murine challenge model, demonstrating its high efficacy in protecting mice from intravaginal challenge with EcPV2 PsVs [80].

VLPs and CVLPs were also assessed as potential tumor-therapeutics in an experimental calf-model (BPV4 VLPs), as well as in naturally sarcoid-affected donkeys and horses (BPV1 CVLPs comprising E7). However, no clear therapeutic effect was observed in these settings [73,91,92]. 

Given the safety and good preventive performance of BPV1 and EcPV2 L1 VLPs in horses and mice, there is legitimate hope that these vaccines will obtain market authorization and become commercially available in the near future.

## Figures and Tables

**Figure 1 viruses-15-00345-f001:**
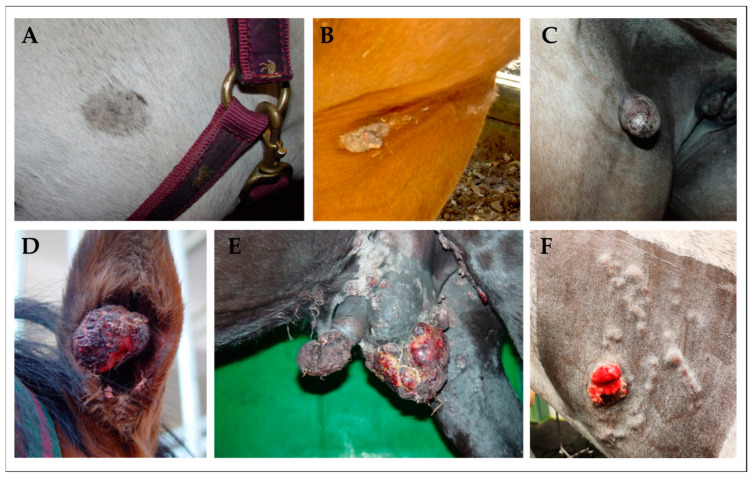
Clinical classification of sarcoids in equids. (**A**): mild-type occult sarcoid presenting as discrete, round-to-oval hairless area on the left jaw of a Warmblood gelding; (**B**): verrucous sarcoid—an irregular, hyperkeratotic area with marked skin-thickening in the axillary region of a Quarter horse; (**C**): nodular sarcoid—a firm dermal nodule covered by apparently intact epidermis in the inguinal region of a Warmblood mare; (**D**): fibroblastic sarcoid affecting the right ear of a Warmblood gelding. The tumor presents as aggressive, ulcerated lesion exhibiting epidermal erosion; (**E**): mixed sarcoid affecting the genital region of a Noriker horse gelding. This tumor conglomerate comprises verrucous, nodular, and fibroblastic lesions; (**F**): malevolent sarcoid on the left neck of an equine patient. This rare, but most aggressive sarcoid-type typically spreads along lymphatic vessels under apparently intact skin, and usually comprises fibroblastic lesions [23].

**Figure 2 viruses-15-00345-f002:**
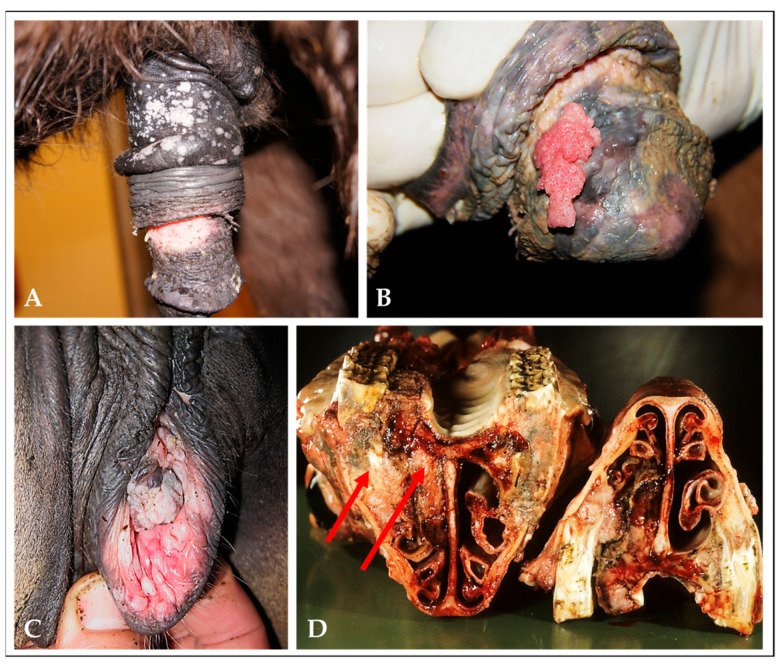
Squamous cell carcinomas (SCCs) in the horse. (**A**) and (**B**): equine-penile-SCC precursor lesions: whitish, benign plaques (**A**) and papillomatous lesion (**B**). (**C**): equine vulvar carcinoma in situ (CIS). (**D**): equine SCC affecting the right maxillary sinus and ventral nasal meatus (red arrows); (this photograph was kindly provided by Dr Andrea Klang).

**Figure 3 viruses-15-00345-f003:**
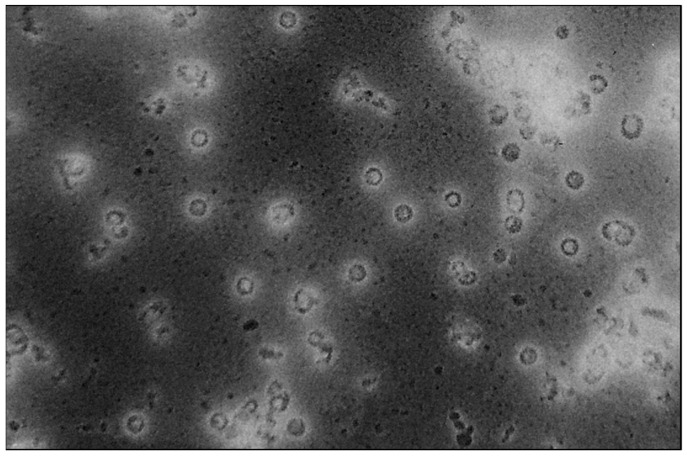
BPV1 L1 virus-like particles (VLPs) used for immunization of horses. Visualization of uranyl acetate-stained BPV1 L1 VLPs by transmission electron microscopy at 30,000× magnification. VLPs were expressed in insect cells and purified on density gradients. This photograph was kindly provided by Saeed Shafti-Keramat from the Medical University of Vienna.

**Figure 4 viruses-15-00345-f004:**
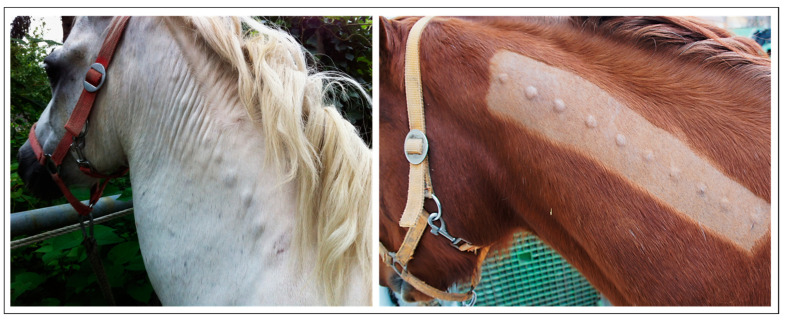
Experimental by bovine δ-PV type 1 (BPV1)-induced pseudo-sarcoids. Intradermal inoculation of horses with confirmedly infectious BPV1/2 virions isolated from cow warts consistently resulted in the development of nodular pseudo-sarcoids, as shown in the examples of a pony (**left**) and a Warmblood gelding (**right**) inoculated at ten sites of the neck. Lesions morphologically and histologically resembled naturally acquired sarcoids but regressed spontaneously.

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
