# Peer review of "Papillomavirus-like Particles in Equine Medicine"

_viruses, 2023, doi:10.3390/v15020345_

Round 1

Reviewer 1 Report

The paper of Edmund K. Hainisch  et al to be published the minor revisions:

Introduction, line 16: this is an incomplete sentence. Recently, ovine papillomaviruses have been shown to be responsible for cross-species transmission and infection (Evidence of a novel cross-species transmission by ovine papillomaviruses. Transbound Emerg Dis 2022; doi: 10.1111/tbed.14756).

Pag. 2, line 1: There are not “semi-benign” tumors, Tumors have benign or malignant biological behaviour. A semi-benign does not exist. Delete, please, “semi”

Pag 2, last two lines: This is true for cattle carcinogenesis, but activation of PDGF beta receptor has never been reported in equids.

Pag 3, line 1: this is an incomplete sentence, Recently, BPVs have been shown to shutdown the immune system via novels pathways. (see Front Immunol 2021, 12;638762; Front Immunol 2022; 13:937795). Both HVs and BPVs evade the immune system via RIG-I-receptors and cGAS/STING pathway.

Pag 3, line 16: BPV13 is now a novel deltapapillomavirus: It is not a geographical variant, Correct it, please.

Finally, in the Conclusion

Line 2: delete, please, semi from semi-benign.

Reviewer 2 Report

the review entitled (Papillomavius-like particles in Equine Medicine) discuss the infection of viruses inducing papilloma in equine and their applications in vaccinology to prevent equine sarcoids and SCC. 

The article is well-written and comprehensive, up-to-date and striking a hot and innovative subject. I am totally recommend the article for publication. 

Author Response

Dear Reviewer, 

we thank you very much for the very positive reception of our article!

Best regards,

Sabine

in the name of all co-authors